# Can Natural Products Targeting EMT Serve as the Future Anticancer Therapeutics?

**DOI:** 10.3390/molecules27227668

**Published:** 2022-11-08

**Authors:** Sirajudheen Anwar, Jonaid Ahmad Malik, Sakeel Ahmed, Verma Abhishek Kameshwar, Jowaher Alanazi, Abdulwahab Alamri, Nafees Ahemad

**Affiliations:** 1Department of Pharmacology and Toxicology, College of Pharmacy, University of Hail, Hail 81422, Saudi Arabia; 2Molecular Diagnostics Unit and Personalized Treatment, University of Hail, Hail 81422, Saudi Arabia; 3Department of Pharmacology and Toxicology, National Institute of Pharmaceutical Education and Research, Guwahati 781101, Assam, India; 4Department of Biomedical Engineering, Indian Institute of Technology Ropar, Rupnagar 140001, Punjab, India; 5Department of Pharmacology and Toxicology, National Institute of Pharmaceutical Education and Research, Ahmedabad 382355, Gujarat, India; 6Department of Pharmacology, Amrita School of Pharmacy, Amrita Vishwa Vidyapeetham, Kochi 641112, Kerala, India; 7School of Pharmacy, Monash University Malaysia, Jalan lagoon Selatan, Bandar Sunway, Petaling Jaya 47500, Selangor DE, Malaysia

**Keywords:** epithelial-mesenchymal transition, cancer EMT, natural chemical entities, cancer, anticancer therapy, chemotherapy

## Abstract

Cancer is the leading cause of death and has remained a big challenge for the scientific community. Because of the growing concerns, new therapeutic regimens are highly demanded to decrease the global burden. Despite advancements in chemotherapy, drug resistance is still a major hurdle to successful treatment. The primary challenge should be identifying and developing appropriate therapeutics for cancer patients to improve their survival. Multiple pathways are dysregulated in cancers, including disturbance in cellular metabolism, cell cycle, apoptosis, or epigenetic alterations. Over the last two decades, natural products have been a major research interest due to their therapeutic potential in various ailments. Natural compounds seem to be an alternative option for cancer management. Natural substances derived from plants and marine sources have been shown to have anti-cancer activity in preclinical settings. They might be proved as a sword to kill cancerous cells. The present review attempted to consolidate the available information on natural compounds derived from plants and marine sources and their anti-cancer potential underlying EMT mechanisms.

## 1. Introduction

Cancer is the leading cause of death globally [1]. According to the global demographic characteristics, it is expected to increase by approximately >20 million by 2025 [2]. The treatment paradigm improved in the past decade with the advancement in cancer research. Breast cancer (BC), colorectal cancer (CRC), lung cancer (LC), and prostate cancer (PC) are the most common types of cancers [1,3]. Various cellular pathways are involved in cancer development and progression. Several drug candidates are approved to target these pathways for their management [4]. One reason behind drug resistance is the process of Epithelial-mesenchymal transition (EMT) involved in cancer progression. EMT is an extremely regulated physiological process that has a significant role in tissue repair and embryogenesis [5]. During EMT, the cells undergo multiple morphologic, biological, and genetic rearrangements, leading to their mesenchymal phenotypes [6]. EMT is pathologically associated with fibrosis and cancer, leading to their progression. EMT has been linked to the formation of invasive and cancer stem cells in carcinomas [7].

EMT is initiated by EMT activating transcription factors (EMT-TFs), including SNAIL (SNAI1) and SLUG (SNAI2), the basic helix–loop–helix factors TWIST1 and TWIST2. As proven for SNAIL, TWIST, Zinc figure E-box binding homeobox 1 (ZEB1), and Zinc figure E-box binding homeobox 2 (ZEB2), these features can repress epithelial genes like the E-cadherin-producing CDH1 by binding to E-Box in their cognate promoter regions. Simultaneously, EMT-TFs activate genes associated with a mesenchymal phenotype, such as Vimentin (VIM), Fibronectin 1 (FN1), and N-Cadherin (CDH2). Several activities, however, are not common and are carried out by separate EMT- transcription factors (TFs) due to differences in coding sequences or protein size and structure [8]. An overview of the EMT pathway is shown in Figure 1. In other words, EMT is a biotic mechanism in which epithelial cells become polarized. 

The heterogeneous mixture of cells like fibroblasts, endothelial cells, noncellular constituents, immune cells, extracellular matrix, cytokines, growth factors, and basement membrane is known as a tumor microenvironment (TME) [9]. EMT is essential for developing and initiating tumors and their recurrence and progression. In TME, the most abundant cells are cancer-associated fibroblasts (CAFs), which cross-talk with tumor cells, extracellular matrix (ECM), immune cells, and endothelial cells for cancer progression [9,10,11,12,13]. Several therapeutic agents are now being designed to target these CAFs [12,14]. Many natural agents have been identified to target CAF by altering the key signaling pathways, epigenetics, kinases, and enzymes. Targeting CAFs and altering the pathways affect cancer-stroma association in TME, resulting in decreased cancer progression. The natural compounds might have promised anti-cancer activity and are worth investigating against different tumors. The characteristic feature of carcinogenesis is ECM stiffness that supports the tumor cells. The crosslinking of ECM components like collagen with the tumor cells occurs via CAFs [15]. LOX-lysyl oxidase, the enzyme highly overexpressed in tumors derived from CAFs, acts as a collagen crosslinking initiator in several cancers like breast and gastric cancers, ultimately enhancing EMT, cell survival, invasion, drug resistance, and angiogenesis [16]. The ECM degrading enzymes like matrix metalloproteases (MMPs) and tissue inhibitors of metalloproteinase (TIMPs) inhibitors are altered by CAFs during angiogenesis and invasion, causing modulation of TME. The MMP2 and 9 are well investigated and highly associated with cancer growth and development [17,18]. The enzymes like metalloproteinases and disintegrin, associated with the MMPs super-family, are increased by CAFs, promoting cancer progression [19]. The CAFs also apply physical forces to pull out the epithelial basement membrane, causing the promotion of EMT in enzyme independent manner [20]. The CAFs promote EMT remolding and are promising therapeutic targets in halting EMT to prevent cancer metastasis. Natural compounds have proven to be the best alternatives to the current therapies against cancer. Many noteworthy examples are in front of us, where natural compounds have proven better than existing standard therapies against different ailments, such as cancer or infectious diseases. It is worth investigating the potential of natural compounds, whether from the marine, plant, or animal, against different types of cancers to find the solution for the growing deadly ailment in the world. Due to growing concerns regarding cancer metastasis, most therapies fail to cure, and the patients suffer greatly due to high toxicity. Cancer resistance is another challenge against current therapies, making cancer more complicated to manage. The natural compounds could be a game changer as anti-cancer therapy that specifically targets the EMT process and halts the process of cancer progression.

## 2. Cross-Talk between TGF-β and Other Signaling Pathways Mediating EMT

The signaling pathways cross-talk to form complex networks. Due to several cellular processes like apoptosis, differentiation, proliferation, and homeostasis, the Transforming growth factor β (TGF-β) cross-talk with various other signaling pathways during the EMT process (Figure 2) [21,22]. One of the mechanisms in which Akt activation and the phosphatase and tensin homolog (PTEN) dissociation from β-catenin are mainly responsible for the TGF-β mediated EMT process, where the displacement of β-catenin from adherent junctions occurs [23]. The other signaling pathway that cross-talks with TGF-β is Notch; Notch synergizes with TGF-β signals to enhance/inhibit its signaling activity depending on the input signal [21]. TGF-β signals activate the migration and inhibit the cell proliferation of endothelial cells. However, the Notch signals block the migration of bone morphogenetic protein (BMP) [24]. BMP stimulates the cell migration of endothelial cells; however, in the presence of Notch signaling, the migratory potential gets inhibited [24].

Interestingly, Notch signaling plays a crucial cross-talk in regulating migration by inducing gene expression. Notch dominates the BMP signaling; when the cell-to-cell contact is not there, endothelial cells are not in contact with the nearby cells to migrate until the new cell-to-cell attachment is set [21,24]. The TGF-β requires Notch signaling for the growth arrest in the epithelium; over thirty percent of the genes induced by TGF-β require Notch signaling [25]. The classy EMT marker, the TGF-β, also cross-talks with several other signaling pathways like Extracellular signal-regulating kinase (Erk), c-Jun N-terminal kinase (JNK), and p38. Erk, JNK, and p38 are indirectly regulating the TGF-β during EMT. However, TGF-β activates MAPK and Erk1/2 signaling pathways [26]. The cross-talk of TGF-β versus EGF signaling is the reason for activating Smad-dependent signaling and MAPK-mediated Erk1/2 [27]. The nuclear translocation of MAPK mediated by TGF-β is downregulated by the MAPK-Erk pathway that mediates nuclear exclusion and phosphorylation of Smad-2/3 [27]. During the initiation of EMT, the Akt and PTEN are also regulated by TGF-β. In addition, TGF-β cross-talk with ErbB signaling during the EMT development of breast cancer [28]. The TGF-β also regulates the phosphoinositide 3-kinase (PI3k)-Akt signaling pathways. Akt’s activity increases due to the induction of TGF-β-mediated functional activities like cell migration, epithelial to mesenchymal shift, cell survival, and cell growth [27,28,29]. Human epidermal growth factor receptor 2 (HER2)/RAS opposes the TGF-β-induced programmed cell death and cell arrest; however, it promotes migratory and invasive activities of TGF-β [30]. The EMT-associated cross-talk is validated by pharmacological inhibition of insulin-like growth factor-1R (IGF-1R), which prevents TGF-β-mediated EMT protein signatures [31]. The cross-talk of different pathways involved in EMT is demonstrated in Figure 2.

**Figure 2 molecules-27-07668-f002:**
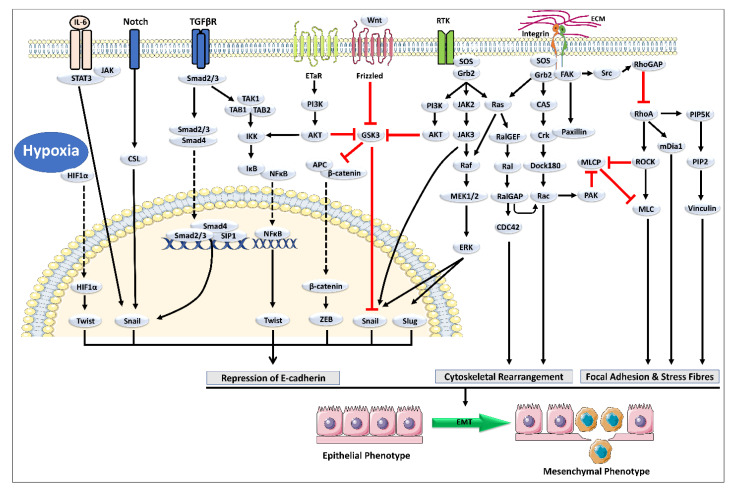
Illustrates the different pathways involved in initiating epithelial-mesenchymal transition (EMT) [32].

## 3. Natural Chemical Agents as Potential Leads against Cancer

Among approved chemotherapeutic medications, 80% are bioactive natural compounds [33,34]. 70% of disease conditions, including cancer, are treated with the help of natural products [35]. The natural compounds induce cytotoxicity by targeting various oncogenic signaling [36]. Several marine-derived metabolites show anti-cancer activity in preclinical and clinical settings [37]. Marine-derived compounds include sulfated polysaccharides, sterols, carotenoids, and chitosan. Sulfated polysaccharides and carotenoids have effectively worked against cancer, acquired immune deficiency syndrome (AIDS), CVDs, and other acute and chronic disorders [35]. Various effective marine-derived compounds are summarized in Table 1. Plants, bacteria, animals, insects, and marine life are some of the key sources of natural chemicals with pharmacological and cytotoxic actions such as anti-proliferative, anti-angiogenic, apoptosis-generating, necrosis-inducing, and anti-inflammatory [38]. Secondary metabolites (Alkaloids, tannins, saponins, flavonoids, steroids) are molecules produced by plants at very minute levels that are therapeutically beneficial in various illnesses [39]. Animals and insects are also rich suppliers of enzymes with various pharmacological applications. Researchers have recently focused on natural substances to investigate their possible use in cancer treatment with lesser side effects. Still, they have only identified a few molecules with clinical efficacy demonstrated in Table 2 [40]. Compounds with anti-proliferative, anti-inflammatory, anti-angiogenic, and apoptotic-inducing properties can treat cancer and reduce the side effects of conventional cancer chemotherapy. They also have a high potential for future use as anti-cancer drugs.

Natural products can offer new hope in fighting EMT and increase therapeutic options worldwide. Drugs of natural origin, like traditional and Chinese medicine, are currently being investigated for various ailments [41]. The cost-effectiveness and richness of good therapeutic efficacy and safety were natural compounds’ true and popular features, considering them promising candidates against cancer [42]. The drugs against cancer at present share 1/3rd position share against cancer, meaning it is important to investigate more drugs from the natural origin against cancer [43]. Natural compounds have a high level of rigidity, enhancing the protein cross-talk more than synthetic drugs. They have diversity and versatile structure complexity, a unique natural feature making them the right candidates against cancer [43,44]. The signaling pathways that are responsible for cancer cell survival and TME maintenance are being halted by many natural compounds. The natural compounds have a comprehensive role in inhibiting tumor progression by blocking the survival pathways involved in EMT [9]. Many drugs have been used along with natural compounds, producing efficacious outcomes like decreasing drug resistance and toxicity [42,45]. Natural products can remold the TME [41]. Here we will show various candidates of natural origin demonstrating pharmacological activities against EMT and its associated factors.

**Table 1 molecules-27-07668-t001:** Marine and plants-derived compounds for anti-cancer activity.

Sr. No	Drug Product	Source (Marine Origin)	Mechanism of Action	Indication	FDA Status	Reference
**Marine Source**						
**1**	Eribulin mesylate	Sponge *Halichondria okadai*	Keeps the cytoskeleton’s growth cycle away from core aggregates tubulin	Metastatic breast cancer	Approved (Spain)	[46]
**2**	Brentuximab Vedotin	Sea hare *Dollabella Auricularia/* cyanobacteria	Cell cycle arrest from G2 to M phase	Hodgkin lymphoma	Approved (USA.)	[47]
**3**	Cytarabine, Ara-C	Sponge *Cryptotheca crypta*	Inhibition of DNA Synthesis	Acute lymphoblastic leukemia	Approved (USA.)	[46]
**4**	Halichondramide (HCA)	Marine sponge *Chondrosia corticata*	Phosphatase of regenerating liver-3 (PRL-3) and its downstream signaling pathway are suppressed.	Prostate Cancer	Approved	[48]
**Plant Source**						
**4**	Ixabepilone	*Soragium cellulosum*	Cell-cycle arrest and apoptosis-inducer	Hand-foot syndrome	Approved	[49]
**5**	Romidepsin	*Chromobacterium violaceum*	Histone deacetylase inhibitors	Hematological toxicities like anemia	Approved	[50]
**6**	Podophyllotoxins	*Podophyllum* (*Berberidaceae*)	Inhibit the polymerization of tubulin, arresting the cell cycle in the metaphase	Ovarian cancer, immunosuppressive ability	Approved	[51]
**7**	Ligustrazine	Rhizome of *Ligusticum wallichii*.	Inhibit SK-OV-3 and OVCAR-3 cell viability, proliferation, migration, and invasion.	Ovarian cancer	Approved	[52]

**Table 2 molecules-27-07668-t002:** Potential NCEs with therapeutic effects against cancer.

Sr. No	NCE.	Source	Mechanism and Outcomes	Method of Validation	Potential Use	Reference
**1**	Oregonin	*Alnus sibirica (AS)*	Anti-proliferative activity, Inhibition of NF-κB, induction of apoptosis, DNA Methylation	MTT Assay, Western blotting, Flow, methylation-specific PCR, cytometry	Prostate cancer	[53]
**2**	Hirsutenone					
**3**	Hirsutanonol					
**4**	Chelerythrine chloride	*Chelidonium majus and Macleaya cordata*	cytotoxicity and anti-proliferative activity	Cell viability assays	NSCLC.	[39]
**5**	Thioholgamide	*Streptomyces* sp. *MUSC 136T.*	Caspase 3/7 Activation, membrane permeability	MTT assay	Colon, breast, liver, and lung cancers	[54]
**6**	7-deoxy-trans-dihydronarciclasin	*Scadoxus pseudocaulus*	Apoptosis inducer	Cytotoxicity assay	Follicular lymphoma	[55]
**7**	4-(4-hydroxy-3-methoxyphenyl) curcumin		Anti-proliferative, apoptosis-inducing	MTT assay, Western blotting analysis	Hepatic, colon, chronic myeloid leukemia, and lung cancer	[56,57,58]

## 4. Potential NCE to Target EMT

### 4.1. Artemisinin (ATM)

ATM is a sesquiterpene lactone isolated from sweet, warm wood, *Artemisia annua.* It is an antimalarial agent to treat multidrug-resistant *falciparum malaria* strains, mediated by producing organic peroxides [59]. ATM also has potent anti-cancer activity against CRC, BC, gastric cancer (GC), and cervix cancer (CC). Its anti-carcinogenic action is similar to antimalarial action in that free iron cleaves its endoperoxide bridge, releasing free radicals that cause cytotoxicity. ATM’s low toxicity and high specificity for cancer cells led to its development as an anti-cancer molecule. Dihydroartemisinic acid (DHA), an ATM derivative, reduced inflammation in a rat arthritis model by downregulating Interleukin-6 (IL-6). Additionally, DHA has anti-cancer properties. It can induce apoptosis in leukemic cells via noxa-mediated mechanisms [60].

Moreover, it inhibits GC cell invasion, migration, and proliferation by inhibiting the activation of phosphoinositide 3-kinase/protein kinase B and SNAIL. According to Sun et al., DHA’s anti-inflammatory and anti-cancer activities are mediated via microRNAs (miRNAs). DHA, for example, inhibits inflammation in vascular smooth muscle cells by regulating the miR-376b-3p/KLF pathway. The Jumonji and AT-rich interactive domain2 (ARID-2)/miR-7/miR-34a pathway inhibit prostate cancer cells by downregulating AXL tyrosine. In laryngeal cancer, miRNAs, in particular, play a function in EMT. For example, miR-217 inhibits EMT while miR-10b promotes it. miR-130b-3p is a tumor suppressor because it inhibits laryngeal cancer development, angiogenesis, migration, and invasion [61]. FoxM1, a member of the conserved forkhead box transcription factor family, is involved in cell cycle regulation, DNA damage repair, and apoptosis and has been associated with the development of breast, pancreas, and liver carcinomas. Nandi et al. hypothesized that FoxM1 was a critical inhibitory target of ATM in hepatocellular carcinoma (HCC) and that FoxM1 may play a role in the cell cycle triggered by DHA ATM-inhibited HCC cell survival and proliferation by attenuating FoxM1 and its transcription targets and interfering with FoxM1 trans-activation [62].

### 4.2. Strychnine/Brucine

Brucine is an alkaloid related to strychnine obtained from the *Strychnic Nux-vomica* tree. It has analgesic, anti-cancer, anti-inflammatory, antioxidant, and anti-venom properties [63]. In vitro, it inhibits the proliferation of Hela and K562 cell lines. Brucine also showed anti-metastasis action in MDA-MB-231 and Hs578-T-cells and inhibited invasive capacity and adhesion of MDA-MB-231 and Hs578-T-cells Matrigel, and preventing mRNA of E-cadherin, catenin, VIM, FN1, MMP-2, and MMP-9 in MDA-MB-231 cells [64]. These data collectively suggest that brucine might be a potential anti-cancer molecule; in vivo studies are still needed to confirm its anti-cancer potential.

### 4.3. Eugenol

A polypropanoid group of compounds is found in seeds of many plants, such as cloves, cinnamon, nutmeg, and bay leaves. It has antioxidant, anti-bacterial, anti-inflammatory, and anti-cancer activity and is widely used as a cosmetic, perfume, and culinary ingredient. It has anti-cancer potential due to its ability to increase reactive oxygen species (ROS) formation and apoptotic action, increase Cyt C’s release, and inhibit the EMT process, limiting the cells’ ability to metastasize [65]. It has shown anti-cancer activity against malignancies, including leukemia, lung, colon, colorectal, skin, gastric, breast, cervical, and prostate cancer, through the processes described below in Table 3.

### 4.4. Resveratrol

Resveratrol (RES) chemically trihydroxy stilbene is a polyphenol in grapes, berries, peanuts, and wine. RES has been shown to have cardioprotective, anti-inflammatory, and anti-aging effects. Studies also suggest that it also has anti-cancer properties. Moreover, RES has a regulatory role in EMT and the hedgehog (Hh) signaling pathway, which is critical for vertebrate development, homeostasis, and cancer. Hh is abnormally activated in breast, prostate, and pancreatic cancer (PC) and has a role in metastasis and invasion of GC via induction of the EMT. Hence, Hh signaling pathway is the center of attraction for anti-cancer activity. RES suppresses the Hh pathway, thus inhibiting cancer invasion and metastasis.

Moreover, RES has also been shown to block the Hh signaling pathway and EMT in malignancies [71]. A study proved that RES inhibited EMT in Glioblastoma (GBM) cells, as evidenced by morphological changes in the RES-treated G.B.M. cells. RES also inhibits EMT-mediated migration and invasion of GBM cells and EMT-induced stem cell-like properties in GBM cells [72]. A TGF-β/Smad signaling pathway is associated with the proliferation, differentiation, and migration of the cells and promotes results in invasion and metastasis. RES inhibited the penetration and metastasis by EMT-induced phosphorylation of Smad2 and Smad3 in a dose-dependent manner, suggesting the function of RES on EMT is related to Smad-dependent signaling [72].

### 4.5. Polyphyllin 1

Polyphyllin 1 (PP 1) demonstrated its anti-cancer activity via its apoptotic action and several pathways effectively against various cancers. Polyphyllin I induces apoptosis in HepF-2-Cells, and neural progenitor cells (NPC) cell lines [73]. The natural herb Paris polyphylla makes PP1 and has anti-cancer properties against various malignancies, including drug-resistant tumors. Paris polyphylla was recently found to inhibit CRC cells by activating autophagy and improving the efficiency of chemotherapy (Doxorubicin). By decreasing CIP2A/PP2A/Akt signaling, PP1 also reduced cisplatin-resistant GC cells. Liu et al. demonstrated that PP 1 has potent anti-cancer action on human non-small cell lung cancer (NSCLC) mediated by CHOP stabilization. PP1 induces ROS, and ER stress inhibits unfolded protein response (UPR) in cancer cells, subsequently increasing the levels of CHOP Via, accelerating CHOP gene expression. The UPR chaperone GRP78, restrained by PP1, is the main mechanism for CHOP stabilization [74].

### 4.6. Paeoniflorin (PF)

Paeoniflorin (PF) is a monoterpene glycoside derived from the root of *Paeonia lactiflora*. In the past, this plant’s roots were utilized in eastern medicine for pain, muscle spasms, inflammation, menstruation dysfunction, and degenerative illnesses for a long time [75,76,77,78]. Studies indicated that PF inhibits tumor growth, invasion, and metastasis in vivo and in vitro. In hypoxia-induced EMT in MDA-MB-231 BC cells, PF treatment resulted in a considerable increase in E-cadherin levels and a drop in CDH2 and Vimentin levels in the cells. Subsequently, it suppressed the EMT process by altering the expression of HIF-1, which is involved in hypoxia-driven EMT [79]. The hippo pathway plays a significant role in the progression of GC. This pathway is said to be dysregulated and thus contributes to gastric oncology and metastasis. Two important factors, yes associated protein (YAP1) and Transcriptional coactivator with PDZ-binding motif (TAZ), produce their metastatic effect via crosslinking with Notch, TGF-β, and Wnt/ β-catenin in GC. In GC, PF exerts its anti-cancer effect via regulation of the hippo signaling pathway and downregulating the effect of TAZ [80]. Further, there is a need to explore its potential in other cancers.

### 4.7. Halicondramine

Halicondramine (HCA) is a trisoxazole-containing macrolide from the marine sponge Chondrosia corticata [81]. It possesses antifungal and cytotoxic properties. It also has anti-proliferative activity against cancerous cells [82,83]. Modulation of the EMT is a key target for their action. Treatment with HCA significantly reduced the expression of MMP2 and 9, and CDH2. On the contrary, E-cadherin expression was significantly increased. HCA also inhibits the expression of PRL-3, a metastasis-associated marker, and PI3 kinase subunits p85 and p110 (PRL-3’s downstream targets). These findings imply that HCA inhibits EMT in human adenocarcinoma prostate cancer cells by modulating PRL-3 and downstream targets, such as PI3 kinase [48].

### 4.8. Ligustrazine

Ligustrazine (LSZ) is obtained from the rhizome of *Ligusticum wallichii* [84]. LSZ is shown to have anti-inflammatory, anti-fibrotic, antioxidant activity, and tumor-suppressing properties in numerous cancers, including LC, GC, BC, and melanoma [85]. LSZ showed anti-proliferative and anti-metastatic action [6]. LSZ increased E-cadherin expression while decreasing the mesenchymal indicators CDH2 and VIM expression. LSZ inhibits EMT in SK-OV-3 cells via modulating miR-211 expression [86].

### 4.9. Fucoidan

The Fucoidan (FC) is a polysaccharide obtained from brown seaweeds and has shown anti-proliferative action on BC cells, such as 4T1 and MDA-MB-231. It also lowered metastatic lung nodules in female Balb/c mice with 4T1 xenografts. The TGFRs molecular network is critical in controlling EMT in cancer cells. It was observed that FC efficiently reverses TGFR-induced EMT morphological alterations, increases epithelial markers, decreases mesenchymal markers and transcriptional repressor expression Twist, Snail, and Slug. Furthermore, fucoidan suppresses migration and invasion during EMT, implying that TGFR-mediated signaling is involved in BC cells [87].

### 4.10. Penisuloxazin A

Penisuloxazin A (PNSA) is a fungal mycotoxin that belongs to a new epipolythiodiketopiperazines (ETPs) possessing a rare 3H-spiro[benzofuran-2,2′-piperazine] ring system. PNSA prevented MDA-MB-231 cell adhesion to coated Matrigels containing several ECM components. Furthermore, after PNSA therapy, there is a transition from spindle-shaped or polygonal mesenchymal to flat polygonal epithelial-like cell morphology. These suggest that PNSA can prevent EMT in MDA-MB-231 cells [88]. PNSA is also considered a potent heat shock protein 90 (HSP90) inhibitor, a well-known N-terminal inhibitor binding to the ATP pocket of HSP90 in preventing BC cell metastasis. Multiple signaling pathways critical for cancer cell proliferation and metastasis can be disrupted by inhibiting HSP90 [89].

### 4.11. Sophocarpine

Sophocarpinr (SC) is one of the most active components of *Sophora alopecuroides* L, a tetracyclic quinolizidine alkaloid. SC has shown various pharmacological actions, including immunoregulatory, anti-inflammatory, and anti-nociceptive [90]. SC has also been shown to preserve heart function from ischemic reperfusion by inhibiting NF-kB and reducing hepatocyte steatosis via activation of the AMPK signaling pathway. Furthermore, in head and neck squamous cell carcinoma (HNSCC) cells, SC has shown anti-proliferative and anti-metastatic by inhibiting dicer-catalyzed miR-21 maturation and activation of the p38MAPK signaling pathway. SC also reduced the HNSCC tumor’s growth in vivo by reversing the EMT in cancer cells. In UM-SCC-22B and UM-SCC-47 cells, SC treatment reduced the expression of the Ki-67 and VIM while increasing E-cadherin’s expression [91]. These findings suggest that SC could be a promising lead drug for HNSCC.

### 4.12. Renieramycin M

Renieramycin M (RM) (22-Boc-Gly-RM), produced by *Xestospongia sp.,* is a semi-synthetic amino ester derivative of the bistetrahydroisoquinoline alkaloid. Studies suggested RM-mediated inhibition of anchorage-independent development and sensitization of detachment-induced cell death in human lung cancer cells [92]. A semi-synthetic derivative of RM with a hydroquinone amino ester extension was synthesized to retain cytotoxicity with increase cancer selectivity [93]. It hinders the phosphorylation of FAK and Akt molecules, which upregulate TIMP2 and TIMP3 and downregulate MMPs expression. The inhibition of the p-FAK/p-Akt signal also marks the downregulation of CDH2 and Rac1-GTP and the upregulation of E-cadherin, where the regulation of cytoskeleton regulatory protein (Rac1-GTP), MMP-associated molecules (TIMP2, TIMP3) [94]. The mechanism of action of RM is demonstrated in Figure 3.

### 4.13. Luteolin

Luteolin (LT) (3,4,5,7-tetrahydroxy flavone) is a flavonoid in many plants including broccoli, carrots, perilla leaves, seeds, and celery. It possesses anti-allergy, anti-inflammatory, anti-cancer, antioxidant, and anti-microbial properties [96,97,98]. In various cancers (including lung, GBM, BC, CRC, PC), LT inhibits cell proliferation and tumor growth, promotes cancer cell apoptosis and cell cycle arrest, reduces drug resistance, and reduces cancer cell invasiveness and metastasis [99]. LT can also stop EMT from occurring, shrinking in the cytoskeleton, increasing the expression of E-cadherin, and decreasing the expression of CDH2, Snail, and VIM [100]. LT inhibits the Smad 2/3 pathway and the Wnt/-catenin pathway by inhibiting the synthesis of Snail and Slug by downregulating the production of β-catenin. Doing so prevents metastasis by upregulating CDH2, Zo 1, and claudin 1 and downregulating CDH2, fibronectin, VIM, and MMP-2 [99,101]. The mechanism of the action of LT is demonstrated in Figure 4.

### 4.14. Carnosic Acid

Carnosic acid (CA), a polyphenolic diterpene found in rosemary (*Rosmarinus officinalis*), has anti-cancer, anti-viral, and anti-inflammatory activities. CA suppresses cancer cell migration and proliferation while lowering vascular endothelial growth factor expression. In leukemia and CRC cells, CA also causes cell cycle arrest at the G2/M phase by downregulating cyclin expression and has been shown to trigger apoptotic cell death in human NB and PrC cells [102]. CA inhibits EMT and cell migration in B16F10 cells in a dose-dependent manner. It prevents Src/AKT phosphorylation and, therefore, activation. It decreases the secretion of uPAc, MMP-9, and TIMP-1, whereas it increases the secretion of TIMP-2 and has no effect on the secretion of MMP-2 and plasminogen activator inhibitor-1 (PAI-1). It is also responsible for the decrease in the expression of Snail and Slug but does not affect the expression of Twist in B16F10 melanoma cells [95]. The mechanism of action of CA is demonstrated in Figure 3 and Figure 4.

### 4.15. N-Phenethylacetamide

N-Phenethylacetamide (NPA) is found in the *Aquamarina Sp.* (MC085). Three compounds, two diketopiperazines [cyclo(L-Pro-L-Leu) (1) and cyclo(L-Pro-L-Ile) (2), and one NPA (3)] isolated with anti-cancer activity. By altering TGF-induced E.M.T., NPA inhibits the TGF/Smad pathway and suppresses A549 cell metastasis. It prevents Snail and Slug expression by inhibiting Smad 2/3 phosphorylation. It also suppresses Snail and Slug, which upregulates the epithelial markers E-cadherin, Zo-1, and claudin-1 while downregulating VIM, FN1, CDH2, and MMP-2 expression, preventing metastasis [101]. The mechanism of NPA is shown in Figure 4.

### 4.16. α-Solanine

α-Solanin (AS), a steroidal glycoalkaloid obtained from nightshade (Solanum nigrum Linn.), suppresses tumor cell growth and causes apoptosis in colon, liver, cervical, lymphoma, and stomach cancer cells. However, the mechanism by which it blocks cancer cell metastasis remains unknown. An animal model of BC induces cell death and inhibits cell proliferation and angiogenesis, resulting in chemotherapeutic actions [103,104]. It also increases E-cadherin expression, reducing VIM expression and cell invasion, which inhibits EMT. It also decreases extracellular inducer of matrix metalloproteinase (EMMPRIN), MMP-2, and MMP-9, increasing Cysteine-rich protein with Kazal motifs (RECK), TIMP-1, and TIMP-2 mRNA expression levels. It downregulated the phosphorylation of Akt, ERK1/2, and PI3K. Furthermore, it increases tumor suppressor miR-138 expression while decreasing oncogenic miR-21 expression [6,105].

### 4.17. Baicalein, Wogonin (WG), and Oroxylin-A (ORA)

Baicalein (BAI), wogonin (WG), and oroxylin-A (ORA) are present in a plant, namely *Scutellaria baicalensis* [105]. It has been reported that the extract of *Scutellaria baicalensis* has anti-tumor activity. A study reported that using the combination of BAI (65.8%), WG (21.2%), and ORA (13.0%) compounds against A549 lung adenoma cancer cells inhibited the EMT process significantly [105]. The Total Flavonoid Aglycones Extract (TFAE) isolated from *Scutellaria baicalensis* has shown inhibition against tumors by inducing apoptosis, mainly BAI, WG, and ORA [106]. It was reported that the TFAE of *Scutellaria baicalensis* has inhibited the EMT of A549 cells via PI3K/AKT-TWIST1 axis [105].

### 4.18. Coptidis Rhizoma

It was reported that the extract of Coptidis Rhizoma (CR) could inhibit the EMT process via the TGF-β signaling pathway [107]. It has been shown that 30% ethanol extract of Coptidis Rhizoma can inhibit cell migration and invasion via blocking E-cadherin and decreasing expression of vimentin, Snail, and ZEB2 [107]. It has a potential anti-metastatic effect and can be a candidate against cancer. The different pathways and proteins targeted by all natural products discussed in the present review are summarized in Figure 5.

## 5. Advantages of Targeting EMT

EMT is recognized as playing a key role in developing cancer, metastasis, and chemotherapy resistance, and its crucial roles throughout cancer progression have recently been discovered and investigated. Although there is still debate about whether EMT causes cancer metastasis, its importance in cancer chemoresistance is becoming more widely recognized, with many EMT-related signaling pathways implicated in cancer cell chemoresistance [39]. Targeted cancer treatments have been an emerging field in the recent decade. Several monoclonal antibody therapies and small chemicals, particularly kinase inhibitors, have been discovered/synthesized and undergo clinical trials with improved anti-cancer effectiveness. While many targeted therapeutic medications demonstrated encouraging preliminary clinical outcomes, such as enhanced overall survival, a significant percentage of patients who received targeted therapy acquired drug resistance following long-term treatment [108]. As a result, cancer drug resistance will determine the success of forthcoming targeted treatment medications. Drug resistance can be caused by various mechanisms, including drug efflux, drug metabolism, and drug target mutations [1,109]. The function of EMT in cancer therapy resistance has recently been explored. In the early 1990s, a relationship between EMT and cancer cell treatment resistance was proposed. Heckford et al. discovered that EMT occurred in two Adriamycin-resistant MCF-7 cells and vinblastine-resistant ZR-75-B cells [110]. Attempts have been devoted to targeting the ABC transporters to overcome drug resistance [111,112]. When it became obvious that EMT plays a critical role in drug resistance, scientists began exploring drugs targeting EMT to overcome drug resistance. Gupta et al. created EMT cells using E-cadherin shRNA and used this cell line to develop CSC-selective small molecule inhibitors. Using high-throughput screening, they discovered an antibiotic named Salinomycin that eliminated breast CSCs preferentially [113]. Salinomycin also reduced EMT caused by doxorubicin exposure and improved doxorubicin sensitivity in HCC cells [114]. It inhibited the expression and operation of drug efflux pumps in BC cells, resulting in a considerable reduction in doxorubicin resistance [115]. In addition to Salinomycin, several minor pharmacologic inhibitors of EMT have been discovered and tested in vitro and in vivo cancer treatment resistance models. Mocetinostat, a histone deacetylase (HDAC) inhibitor that restored miR-203 and decreased ZEB1 (EMT-TF) expression, reversed EMT in drug-resistant cancer cells and sensitized them to the chemotherapeutic agent docetaxel [116]. Curcumin, a component of curry, was discovered to sensitize 5-fluorouracil-resistant colorectal cancer cells via inhibiting EMT via miRNA [117]. According to Namba et al., EMT mediated by the Akt/GSK3/Snail1 pathway was a critical signaling event in acquiring gemcitabine resistance in PC cells. The anti-viral zidovudine inhibited these signaling pathways, restoring gemcitabine sensitivity in cancer cells. Co-administered zidovudine with gemcitabine reduced tumor growth and prevented cancer cells from establishing the EMT phenotype in mice with a gemcitabine-resistant pancreatic tumor xenograft [118]. Oncologists have recently focused on metformin since it has anti-cancer and chemopreventive qualities independent of anti-hyperglycemic effects [119,120]. Hirsch et al. later discovered that metformin targets BCSCs [121]. According to follow-up studies, metformin lowers CSCs by targeting EMT Metformin triggered transcriptional re-programming of BCSCs by lowering major EMT-TFs such as SNAIL2, Twist1, and ZEB1, according to Vazquez-Martin and colleagues [121]. Metformin has been shown to prevent EMT in lung cancer by inhibiting the IL-6/STAT3 axis in lung adenocarcinoma [122]. Although the direct molecular target of metformin in suppressing EMT is unknown, the Stimulation of AMPK may play a significant role in the drug’s anti-EMT activity [123,124]. Metformin is being studied in over 200 human clinical studies for cancer therapy because of its possible CSC, anti-cancer properties, and favorable safety profile [125]. As a result, targeting EMT has been viewed as a unique strategy for combating cancer treatment resistance. In addition to the small compounds that have been created, a lot of pharmacological screening is being done to find new EMT inhibitors. Chua et al. created an EMT spot migration recognition method that can be utilized for high-content screening to screen small molecule EMT inhibitors that target certain growth factors. Scientists could undertake high throughput screening of small compounds utilizing enhanced screening platforms thanks to advancements in EMT and CSC biology [126]. Aref et al. also created a microfluidic device that mimics the 3D tumor microenvironment by including tumor cell spheroids and an adjacent endothelial monolayer. This approach is very beneficial in identifying EMT medicines active in a complex in vivo tumor microenvironment with several cell kinds interacting [127,128]. As a result, targeting EMTs with the natural chemical entity is viewed as a unique and innovative approach to combating cancer treatment resistance.

## 6. Targeting EMT Process by Molecular Docking (MD)

MD is a promising approach for estimating the interaction between biological molecules, such as proteins and ligands [129,130]. In the last decade, MD has emerged as a promising tool in identifying lead molecules against different ailments. MD is applied to cancer stem cells (CSC) associated metabolic and signaling pathways. Different metabolic pathways participate in CSC survival concerning cancer progression and alterations [130]. This is why metabolic re-programming is considered one of the cancer symbols [131].

Natural products are considered the grounds of multi-targeting molecules. Alkaloids, a class of natural producers, are one of the promising molecules that have the strength to combat CSCs via MD [130,131]. The drugs like salasonine and tylophorine have shown that they altered the Hedgehog (Hh) pathways and exerted anti-cancer effects on CSCs [130,131]. The targeting of overexpressed receptors in cancer tissues has demonstrated anti-cancer potential by natural products earlier through MD and experimental approaches [130,131]. It is reported that natural products represent a rich source of therapeutically active compounds which can interact with numerous cellular targets and minimize the side effects [132]. MD may help us find the natural lead molecules against the EMT process. Different receptors play a role in the EMT process, so it will be more important to screen out natural compounds against those receptors to find lead molecules.

## 7. Future Prospective

EMT has long been suspected of contributing to cancer therapy resistance. It became clear when scientists observed strong parallels in gene expression profiles and marker expression between EMT cells and CSCs. The resistance of CSCs against pharmacotherapy is a major challenge and poses a significant threat to cancer patients. The pathways involved in EMT have been deciphered by scientists, and have developed several methodologies to investigate the phenotypes in EMT. These understandings form the basis for drug screening against EMT-mediated cancer. The miRNA and some chemical agents demonstrate inhibitory activity against EMT, but no currently available miRNAs in clinical settings can solve this problem. So natural compounds showing good results can be the game changers in mitigating the EMT process. Several small chemical agents have been discovered to help drug-resistant cancer cells targeting EMT become more chemo-sensitive. Many of these, including Ligustrazine Penisuloxazin A, Halichondramide, Sophocarpine, Fucoidan, and Diketopiperazines, were investigated in human clinical trials with standard chemotherapies or targeted treatments.

Furthermore, EMT inhibitors’ long-term safety is unknown. This is especially true if EMT inhibitors activate the MET pathway, which has been linked to cancer metastasis. Moreover, studies need to be conducted to investigate the link of EMT with different pathways involved in Cancers. There is a need to develop nanoformulations targeting the cancerous to avoid off-target side effects. Moreover, with the advancement of the compounds screening assays, more novel natural compounds need to be explored to find a more potent EMT inhibitor. Novel in vitro and in vivo approaches should be developed to reduce translational failure [133].

## 8. Conclusions

The part of EMT-MET in cancer cell dispersion and distant metastasis has been acknowledged. Recent research suggests that EMT may not be required for cancer cell metastasis despite its importance in chemoresistance. However, this is controversial because of the EMT phenotype’s variability and adaptability. Only a fraction of EMT populations may be controlled by Fsp1. Finally, EMT is a critical cancer cell characteristic contributing to medication resistance. Inhibitors of this biological mechanism will be suitable “partners” for chemotherapy or targeted therapy medications, allowing current cancer therapies to achieve better clinical outcomes. Natural products play a dynamic role in controlling the process of EMT in cancer; many assays have been performed to prove their activity. EMT is a progression where cells can lose their epithelial properties like E-cadherin and a few more and gain mesenchymal properties like CDH2, VIM, FN1, etc. EMT has long been investigated for its involvement in cancer treatment resistance and metastasis. Assay techniques to analyze EMT phenotypic and drug screening have been created based on a better knowledge of natural chemicals and critical signaling pathways in EMT. Natural compounds that downregulate EMT phenotype and, as a result, drug resistance and metastasis have been identified, allowing scientists to discover natural compounds as EMT inhibitors capable of improving chemosensitivity of drug-resistant cancer cells while inhibiting metastasis. However, additional research is needed to fully comprehend the significance of EMT in cancer treatment resistance, cell proliferation, invasion, metastasis, and natural chemicals and their involvement in blocking EMT.

## Figures and Tables

**Figure 1 molecules-27-07668-f001:**
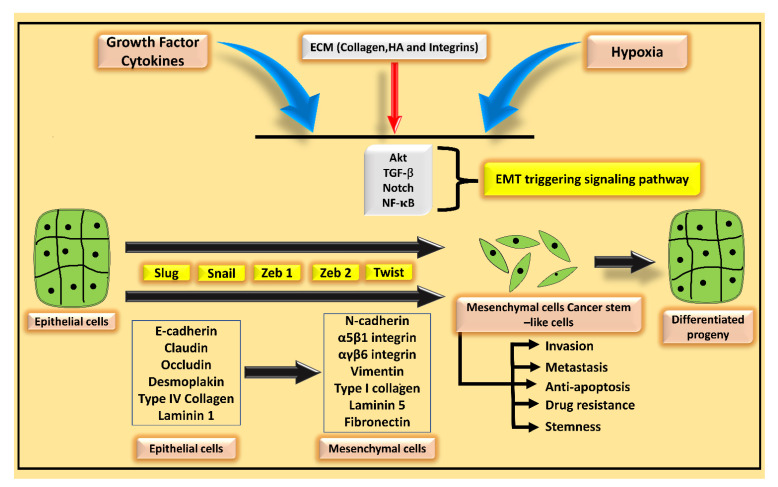
Hypoxia, growth factor, cytokines, and ECM activate the pathways that can trigger the EMT by activating EMT transcription factors (EMT-TFs), including SNAIL, SLUG, and the basic helix–loop–helix factors TWIST. As proven for SNAIL, TWIST, Zinc figure E-box binding homeobox 1 (ZEB1), and ZEB2, these features can repress epithelial genes like the E-cadherin-producing CDH1 by binding to E-Box in their cognate promoter regions. Simultaneously, EMT-TFs activate genes associated with a mesenchymal phenotype, such as Vimentin (VIM), Fibronectin 1 (FN1), and N-Cadherin (CDH2), etc. [8].

**Figure 3 molecules-27-07668-f003:**
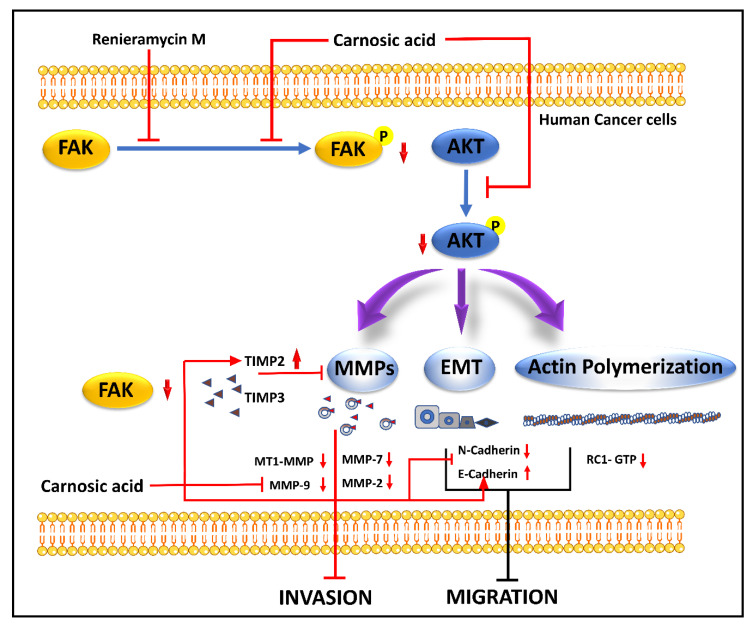
Inhibition of the FAK/AKT signaling pathway and subsequently decrease EMT markers, CDH2 MMPs, etc., and increases epithelial marker, E-cadherin, which reduces cell invasion and migration by Renieramycin M and Carnosic Acid [94,95].

**Figure 4 molecules-27-07668-f004:**
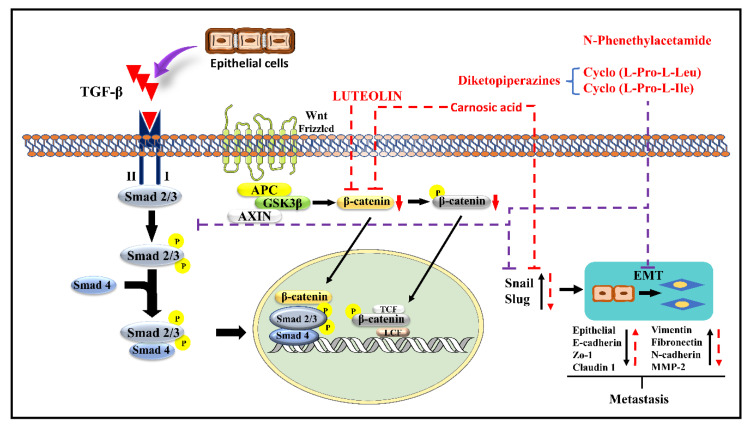
Possible mechanisms of luteolin, Carnosic acid, and N-Phenethylacetamide inhibit EMT via different pathways [94,95].

**Figure 5 molecules-27-07668-f005:**
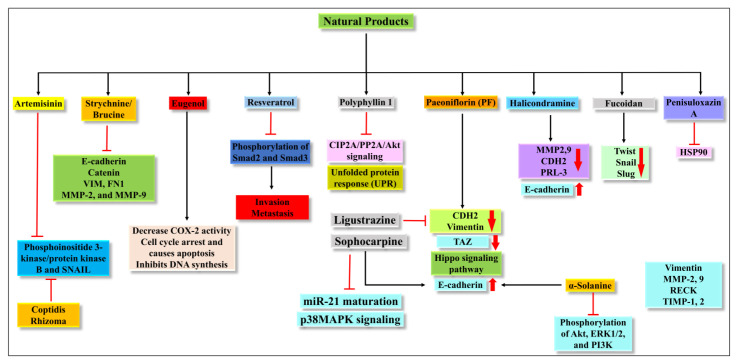
Illustration of the different pathways targeted by the natural products discussed in the present review.

**Table 3 molecules-27-07668-t003:** Mechanisms of Eugenol for anti-apoptotic in various cancers.

Type of the Tumor	Study Type	Effective Dose	Mechanism	References
Lung cancer	In vitro	1000 μM	Decrease cycloxygenase-2 activity, which leads to cell cycle arrest in the S phase followed by cell death	[66]
Colon cancer	In vitro	800 μM	Boosts the cytotoxic effects of cisplatin and doxorubicin synergistically.	[67]
Gastric cancer	In vitro	Low conc.	Inhibits cancer growth by upregulating preinvasive and angiogenic molecules and favoring apoptosis via the mitochondrial pathway via altering Bcl-2 family proteins.	[68]
Cervical cancer	In vitro	50–200 μM	Prevents the cell cycle and causes apoptosis, and inhibits DNA synthesis.	[69]
Breast cancer	In vitro	2 μM	Suppresses breast cancer-related oncogenes by downregulating E2F1 and its downstream anti-apoptotic target	[70]

## Data Availability

Not applicable.

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
