# Peer review of "Can Natural Products Targeting EMT Serve as the Future Anticancer Therapeutics?"

_molecules, 2022, doi:10.3390/molecules27227668_

Round 1

Reviewer 1 Report

molecules-1981233 titled "Can natural products serve as anticancer lead compounds that target the epithelial-mesenchymal transition pathway?" is described the natural products were summarized for their molecular mechanism associated with the EMT pathway. This manuscript could provide information for the reader and be valuable. 

Before acceptance, several points should be modified. 

1) Numbering of Subtitles: 1. Introduction- then, line 123; 2. Cross-talk ~, line 193 3.; EMT~, line 211, a. EMT in diabetes, line 222, b. EMT in ~ , line 240 4. natural chemical~

2) Regrouping: Table 2, 8  Halichondramide (HCA) >> 4. (HCA is obtained from marine origin)

3) Page 8, line 499: Mcf-7 >> MCF-7 cells 

Author Response

Molecules-1981233 titled "Can natural products serve as anticancer lead compounds that target the epithelial-mesenchymal transition pathway?" is described the natural products were summarized for their molecular mechanism associated with the EMT pathway. This manuscript could provide information for the reader and be valuable. 

Authors Response: We thank the reviewer for the positive response towards our research.

Before acceptance, several points should be modified. 

  • Numbering of Subtitles: 1. Introduction- then, line 123; 2. Cross-talk ~, line 193 3.; EMT~, line 211, a. EMT in diabetes, line 222, b. EMT in ~ , line 240 4. natural chemical~

Authors Response: We thank the reviewer for improving our article and all suggested changes are implemented.

  • Regrouping: Table 2, 8  Halichondramide (HCA) >> 4. (HCA is obtained from marine origin)

Authors Response: We thank the reviewer for improving our article and all suggested changes are implemented.

  • Page 8, line 499: Mcf-7 >> MCF-7 cells 

Authors Response: We thank the reviewer for improving our article and all suggested changes are implemented.

Reviewer 2 Report

This review focused on the natural compounds derived from plants and marine sources that possess their anticancer activities through targeting EMT. 

Major points:

1. Since the focus of this review was on cancer, the "EMT and other pathological disorders" section and Figure 3 should be removed from the manuscript. 

2. Either Figure 1 or Figure 2 should be revised to illustrate effects of representative natural products on individual signal pathways or effector molecules that mediate EMT. 

3. There was a whole section reviewing information on "Cross-talk between TGF-β and Other Signaling Pathways Mediating EMT". However, of the 18 natural products reviewed, only three compounds exhibit inhibitory effect on EMT through the TGF-β related mechanism. Have the authors done a thorough literature search to see if any other natural products have any direct or indirect effect on the TGF-β mediated EMT?

Minor point:

The numbering of section headings was messed up. 

Author Response

This review focused on the natural compounds derived from plants and marine sources that possess their anticancer activities through targeting EMT. 

Major points:

  1. Since the focus of this review was on cancer, the "EMT and other pathological disorders" section and Figure 3 should be removed from the manuscript. 

Authors Response: We thank the reviewer for improving our article and all suggested changes are implemented.

  1. Either Figure 1 or Figure 2 should be revised to illustrate effects of representative natural products on individual signal pathways or effector molecules that mediate EMT. 

Authors Response: We thank the reviewer for improving our article and all suggested changes are implemented.

  1. There was a whole section reviewing information on "Cross-talk between TGF-β and Other Signaling Pathways Mediating EMT". However, of the 18 natural products reviewed, only three compounds exhibit inhibitory effect on EMT through the TGF-β related mechanism. Have the authors done a thorough literature search to see if any other natural products have any direct or indirect effect on the TGF-β mediated EMT?

Authors Response: We thank the reviewer for improving our article and all suggested changes are implemented.

Minor point:

The numbering of section headings was messed up. 

Authors Response: We thank the reviewer for improving our article and we have rectified numberings and headings.

Reviewer 3 Report

This is a loosely constructed review article and need serious re-write. The flow of introduction is not up to the mark and must be revised according to the theme of paper. 

Authors constructed this paper very casually as they used two different template such as Molecules and Cancers which is not appreciable. I would suggest careful proofread of the paper before uploading. 

References are not uniform and recent references are less. 

Please see paper for further comments. 

Author Response

Reviewer 3

This is a loosely constructed review article and need serious re-write. The flow of introduction is not up to the mark and must be revised according to the theme of paper. 

Authors constructed this paper very casually as they used two different template such as Molecules and Cancers which is not appreciable. I would suggest careful proofread of the paper before uploading. 

References are not uniform and recent references are less. 

Please see paper for further comments. 

Authors Response: We thank the reviewer for improving our article and we have rectified all points raised by the reviewer. All changes are highlighted with yellow colour.

Round 2

Reviewer 2 Report

The authors have addressed all my concerns.

Reviewer 3 Report

ALL CEOMMENTS HAS BEEN RECTIFIED.